# Integrating Patient-Specific Information into Logic Models of Complex Diseases: Application to Acute Myeloid Leukemia

**DOI:** 10.3390/jpm11020117

**Published:** 2021-02-10

**Authors:** Alessandro Palma, Marta Iannuccelli, Ilaria Rozzo, Luana Licata, Livia Perfetto, Giorgia Massacci, Luisa Castagnoli, Gianni Cesareni, Francesca Sacco

**Affiliations:** 1Department of Biology, University of Rome Tor Vergata, Via delle Ricerca Scientifica 1, 00133 Rome, Italy; alessandro.palma@live.it (A.P.); marta.iannuccelli@gmail.com (M.I.); ilaria.rozzo@gmail.com (I.R.); luana.licata@gmail.com (L.L.); livia.perfetto@gmail.com (L.P.); giorgiamassacci@hotmail.it (G.M.); Castagnoli@uniroma2.it (L.C.); 2Fondazione Human Technopole, Via Cristina Belgioioso, 171, 20157 Milan, Italy

**Keywords:** Boolean networks, logic modelling, acute myeloid leukemia, signaling

## Abstract

High throughput technologies such as deep sequencing and proteomics are increasingly becoming mainstream in clinical practice and support diagnosis and patient stratification. Developing computational models that recapitulate cell physiology and its perturbations in disease is a required step to help with the interpretation of results of high content experiments and to devise personalized treatments. As complete cell-models are difficult to achieve, given limited experimental information and insurmountable computational problems, approximate approaches should be considered. We present here a general approach to modeling complex diseases by embedding patient-specific genomics data into actionable logic models that take into account prior knowledge. We apply the strategy to acute myeloid leukemia (AML) and assemble a network of logical relationships linking most of the genes that are found frequently mutated in AML patients. We derive Boolean models from this network and we show that by priming the model with genomic data we can infer relevant patient-specific clinical features. Here we propose that the integration of literature-derived causal networks with patient-specific data should be explored to help bedside decisions.

## 1. Introduction

Some diseases are caused by mutations in a single gene, or in a few genes [1]. Understanding the function(s) of the mutated gene(s) can shed light on the molecular mechanisms that are altered in the disease condition. At least in principle, therapeutic interventions can be rationally planned by negatively or positively targeting the molecular path influenced by the disease gene(s) [2]. More often, however, whole genome association studies of large cohorts of patients reveal the association of a specific medical condition to variants in a large number of loci, each contributing to a variable quota for the disease manifestation [3]. No single gene can be held responsible for the condition and many genes cooperate to modify the functioning of a large network governing a complex physiological function. In this latter case understanding how the disease genes cooperate and delineating the connections of the underlying molecular networks, although essential, often turns out to be a daunting task. In this case tracing therapeutic strategies requires a complete quantitative understanding of the mechanisms by which the disease genes talk to each other and of how these interactions, or lack of them, induce a perturbation of the network causing the disease.

AML is a life-threatening, complex and heterogeneous disease caused by the uncontrolled expansion of myeloid precursors [4]. Exon and whole genome sequencing studies have revealed genetic alterations in multiple genes, thus providing a wealth of genetic information, much of which remains to be interpreted mechanistically. Gilliland and Griffin have observed some regularities in the mutational profiles of AML patients and have proposed a two-hit model [5]. According to this model, most AML cases are largely caused by the co-occurrence of driver mutations that can be grouped into two different classes. Class I mutations occur in genes (such as FLT3, NRAS, PTPN11) involved in signal transduction and conferring a proliferative advantage, while class II mutations hit epigenetic modulators or transcription factors (DNMT3A, CEBPA, CREBBP, EP300) impacting hematopoietic differentiation. This model enabled, to some degree, the rationalization of the mutational path and the molecular mechanisms causing AML, a disease triggered by a differentiation block and uncontrolled proliferation. Although useful, this model does not comprehensively explain all AML cases. As the number of fully characterized patient genomes has dramatically increased, it has become clear that often patients are mutated in genes that do not fit into one of the two previously described classes [6] and that AML is a dynamic disease, characterized by multiple competing clones evolving over time and carrying diverse infrequent mutations [7]. Systematic collaborative projects aimed at the characterization of the mutational landscape of AML patients have generated a large catalogue of leukemia genes that is increasingly comprehensive [8,9]. Importantly, it was shown that the prognostic value of individual mutations is often significantly altered by the co-occurrence of additional driver mutations. An increasing number of studies have already successfully combined tumor profiles with network-based approaches to stratify patients or to obtain cancer pathway maps [10].

Given the increasing availability of genome-scale mutational profiles, we propose a novel network-based approach combining a literature-derived AML causal network with patient-specific genetic profiles. In our strategy, we apply Boolean logic to investigate in silico some of the somatic mutations on the activation level of hallmark processes and predict clinical outcomes. Our work aims at developing strategies to exploit large-scale patient genomic information to infer clinical outcomes and design effective personalized therapy.

## 2. Materials and Methods

### 2.1. Curation of AML-Relevant Causal Information

To assemble a network underlying the molecular mechanisms that are disrupted in the different forms of AML we first defined a list of “AML driver genes” by surveying the genes frequently mutated in AML as listed by four international sequencing consortia or online resources [8,9,11,12]. The 31 genes mentioned in at least two of the four lists were considered. We also considered as “AML driver genes” MYC, ETV6 and CBFB as they are highly interconnected in the AML network. Finally, we added the six fusion proteins (AML1-ETO, BCR-ABL, CBFbeta-MYH11, MLL-fusions, PML-RARalpha NUP98 fusion proteins) caused by genome rearrangements, as they are frequently observed in AML patients. In a second step, we performed a literature search by standard methods (PubMed and Google searches) looking for scientific reports describing the molecular mechanisms or causal relationships connecting driver genes to AML hallmark phenotypes (apoptosis, differentiation and proliferation). Relevant information was manually annotated in SIGNOR according to the database curation rules [11]. 

### 2.2. Assembly and Curation of a Network Linking Driver Genes and Cancer Hallmarks by Causal Relationships

The dedicated curation effort described in the previous section captured results of experiments in a total of 81 articles and resulted in the annotation of 200 new AML-related causal relationships. This information, available at https://signor.uniroma2.it/, was used to connect the driver genes and the hallmark phenotypes via causal relationships. To this end we took advantage of CancerGeneNet [12], a resource designed to investigate signaling paths between any gene and cancer hallmarks. We used a tool offered by the resource to link the 40 AML driver genes and the cancer hallmarks’ proliferation, differentiation and apoptosis. The tool returned a network of 517 nodes and 2508 edges which was first filtered to remove genes that are not expressed in leukemia cell lines and/or patients (Figure A1), according to previously published proteomic and transcriptomic AML datasets [9,13]. Next, the network was further refined by expert curators to yield a graph of 81 nodes and 130 edges, dubbed the “AML network”. For Boolean simulations aimed at investigating the crosstalk between genes that are frequently comutated in patients, we extracted a simpler network of approximately 30 nodes (AML modules), which connects the relevant driver genes to hallmark phenotypes. All the networks and modules described in this report can be downloaded from the module section of the CancerGeneNet database (https://signor.uniroma2.it/CancerGeneNet/).

### 2.3. Developing Boolean Models from Logic Networks

Boolean models were derived from the logical networks by associating a Boolean rule to each node of the network (Table A1) using the Boolean operators “AND,” “OR” and “NOT” or a combination of them in an effort to best describe the activation logic of the node according to available experimental evidence. Whenever two or more activating or inhibiting interactions converged to a single node they were put in “OR” with exception for known “AND” constraints. Thus, each node of the network is associated to a Boolean expression, describing how the value of the node changes depending on the activities of the upstream regulatory nodes. Whenever a node receives more than one input it is necessary to have information on the combined effect of the multiple signals in order to implement logic gates governed by Boolean operators. Most of the times this information is, however, not available. To overcome this stumbling block, in the absence of experimental information, we used the following arbitrary “inhibitor wins” approach for combining the effect of different inputs [14]. Two inputs of the same sign are linked by an “OR” operator while, in the few cases where a node receives inputs of different sign, we assume that inhibitions wins over activations. The state of the network is described by a vector of zero and one values each representing the activation of a node in the network. The simulation ends when the network reaches a stable configuration, remaining unchanged over time. We used the R package Boolnet [15] to assemble the model and to compute its steady states. These network configurations correspond to stable patterns of expression that can be reasonably associated to biological states. Nodes corresponding to loss of function mutations were set to 0 and were not changed during simulation irrespective of the activation states of the upstream nodes (in silico knockout). Constitutive oncogene expression is simulated by fixing to 1 the value of the corresponding node. 

## 3. Results

### 3.1. A Network-Based Strategy

We aim to develop a generally applicable approach to improve inference of patient clinical outcomes from genomic data. Recently, different network approaches have been developed, already providing some value in patient diagnosis and prognosis [11,12,13]. Here we reason that imposing an additional level of information about gene and gene products’ causal interactions, including directionality and sign, might help to improve inference of disease (here AML) clinical outcomes from patient genomic data. 

To this end we developed a four step-strategy that can be generally applied to complex diseases (Figure 1).

Identification of driver genes. First, we took advantage of different disease mutation databases to annotate as disease genes those genes that are found frequently mutated in patients diagnosed with the specific pathology.Connecting driver genes to hallmark processes. Next, we used CancerGeneNet, a tool implemented in the resource SIGNOR, to connect the disease genes to hallmark disease phenotypes by causal relationships, obtaining a naïve network of cause–effect interactions.Generation of pathway modules. This large network is broken down into smaller modules representing functional path detailing how the most common co-occurring mutations may functionally interact to regulate hallmark phenotypes.Development of disease-specific Boolean network. The logic information underlying the module network-topology is translated into actionable Boolean models to be used to infer the combined effect of the different mutations on patient prognosis.

In the next paragraphs we will describe how we applied this strategy to obtain AML predictive logic models. 

### 3.2. Identification of AML Driver Genes

As a first step we used available information on genes frequently mutated in AML patients to define AML driver genes. Several collaborative studies have reported the genomic characterization of AML patients [8,9,14,15]. These data are annotated in a number of resources and publications [16]. We screened the literature and online resources to capture and integrate these data in order to obtain a comprehensive and reliable list of genes whose genetic alteration causes AML. We integrated the information contained in four independent gene lists:The TCGA AML dataset, consisting of genomic data of 200 clinically annotated adult cases of de novo AML patients. Fifty cases were characterized by whole-genome sequencing while for the remaining 150 only the exomes were sequenced. This study, which is one of the sequencing projects of the landmark cancer genomics program TCGA, enabled the identification of 23 significantly and recurrently mutated genes [9].Papaemmanuil et al., NEJM 2016, reported the mutational profile of 111 cancer driver genes in 1540 patients enrolled in three clinical trials. For each patient cytogenetic and clinical data are also available [8].Cancer Gene Census, a continuously updated resource storing a manually expert-annotated catalogue of genes containing mutations that have been causally implicated in cancer onset or progression [16].DisGeNET, a resource containing one of the largest collections of gene lists associated with human diseases [17]. The list of genes that this resource links to AML is very large (102 genes). We only considered the 33 genes significantly associated to AML (Score_dga > 0.02).

Perhaps surprisingly, we observed little overlap between these four lists (Figure 2a). This could be explained by taking into consideration that the two clinical studies only investigate adult cases of de novo AML, while both Cancer Gene Census and DisGeNET list genes mutated in both pediatric and adult AML patients in both de novo and therapy-related AML. 

Additionally, while Papaemmanuil et al. sequenced a subset of about 100 selected cancer genes, in the TCGA study whole exome sequencing is performed to obtain an unbiased list of mutated genes. Finally, Papaemmanuil et al. did not consider fusion proteins, which are present in the other lists. 

Thus, the four datasets were generated by applying different experimental strategies to heterogeneous cohorts. However, considering as drivers all the genes annotated in the four independent resources has the advantage of being a comprehensive approach, at the cost of risking the inclusion of false positives. This would generate a very large network whose dynamic behavior might be too complex to interpret. On the other hand, considering only driver genes that are annotated by two or more resources would result in a more reliable yet smaller network.

For the scope of this analysis, we adopted a more conservative approach and considered 31 genes as AML drivers, selecting only those genes that were annotated as drivers in at least two out of the four resources (Figure 2b). In addition, MYC, ETV6 and CBFB were retained in the list as they are highly connected with the other driver genes [8].

In addition, we have also included in the AML driver-gene list the six fusion proteins most commonly identified in AML patients: AML1-ETO, BCR-ABL, CBFbeta-MYH11, MLL-fusions and PML-RARalpha and those involving the NUP98 gene, as about 20% of AML patients show chromosomal translocations, which often cause gene fusions encoding onco-fusion proteins [18].

### 3.3. Connecting AML Driver Genes to Hallmark Processes

Next, we aimed at connecting, via causal relationships, the 40 “AML driver genes” and linking them to cancer hallmarks. Although AML subtypes can have rather different and recognizable morphological and clinical manifestations, they are all characterized by a hematopoietic differentiation-block accompanied by uncontrolled replication of nondifferentiated cells. Thus, we reasoned that the different observed AML driver mutations, although sometimes apparently unconnected, would contribute to modulate the functioning of a large connected network affecting the three cancer hallmarks’ phenotypes, differentiation, proliferation and apoptosis. 

Briefly, using the causal relationships annotated in the SIGNOR database [11] we generated a large network that aims at describing how AML cancer driver genes talk to each other and may interact to modulate the disease phenotypes. Differently from other network approaches that are based on binary physical interactions [19,20,21], the network that we generated is represented as a signed directed graph of cause–effect relationships that can be used to draw actionable Boolean models. 

To generate such a network, we exploited CancerGeneNet. However, a preliminary evaluation showed that only 20 of the 40 AML driver genes were annotated with causal relationships in the resource. To fill this gap, we embarked on a literature curation effort to link driver genes to the hallmark phenotypes “proliferation”, “differentiation” and “apoptosis” by capturing relevant experimental evidence. 

The resulting network was filtered to include only proteins expressed in leukemic cells. This information was gathered from the tissue-specific transcriptome and proteome profiles of leukemia patients and AML cell lines. This approach yielded an AML-specific directed network of 111 nodes and 517 edges (Figure A1). This automatically generated network was further reviewed and pruned, taking into account expert consensus as extrapolated from highly cited reviews [22,23] to generate a simpler graph of 81 nodes, 40 of which are AML driver genes, and 130 edges (Figure 3a). As the information used to generate the network is causal, the edges linking the proteins in the network have a direction and a sign that are represented with arrow or T shaped edges and colors in Figure 4a.

The procedure that we used to generate the AML specific causal network, although general, is affected by a degree of arbitrariness. As a test of the self-consistency and the predictive value of the assembled AML network, we assessed whether the network topology and the signs of the graph edges would allow us to correctly classify the network nodes representing driver genes as oncogenes or tumor suppressors. To this end we defined as oncogenes those genes stimulating cell proliferation and/or inhibiting differentiation or apoptosis, while the genes activating differentiation or apoptosis and/or suppressing proliferation were classified as onco-suppressors. This approach enabled us to classify 18 genes as oncogenes and 19 as tumor suppressors (Figure 3b). We found that three additional genes were connected to hallmark phenotypes by different paths that were compatible with an oncogenic or a tumor suppressor potential depending on the signaling path. We next compared our automatic unbiased classification with that provided by the experts of the Cancer Gene Census. Remarkably, 24 genes obtained the same classification by both approaches. The Cancer Gene Census annotates nine genes as both oncogenes and onco-suppressors depending on the context and tissue specificity [24,25,26,27]. Thanks to our AML-specific network approach, we were able to assign an unambiguous classification to these nine genes: eight of these had paths that were compatible with tumor suppressor activity, one with an oncogenic potential.

### 3.4. AML Modules 

Causal networks can be relatively easily converted into actionable Boolean models provided that experimental evidence is available to implement AND/OR operators into logic gates. Boolean models, although somewhat simplistic, have been shown to be valuable in capturing the basic properties of a biological system and its dynamics [16,17,18,19,20,21,22]. Each node of the network is associated to a Boolean expression, describing how the value of the node changes depending on the activities of the upstream regulatory nodes. Equilibria of the system may be often associated to a specific cell phenotype. An additional advantage of Boolean models is their scalability. However, Boolean simulations of large networks are not only computationally challenging, but also generate complex results, often offering unclear biological insights.

The AML network that we generated, consisting of 81 nodes connected by 130 edges, is large and complex and as such of limited practicality. Thus, we aimed at simplifying it by extracting different independent network modules illustrating how the most frequent pairs or triplets of co-occurring mutations impact the cancer hallmark phenotypes’ proliferation, differentiation and apoptosis. Thus, we generated four network modules representing the molecular mechanisms connecting the gene products that are frequently comutated in AML patients (Figure 4). These simplified pathways consist of 20–30 nodes connected by about 35 edges, and are more practical for model simulations.

As shown in Figure 5, these models are still generic as they do not include any patient-specific information. Additionally, these simplified models are assembled by integrating incomplete experimental evidence and are somewhat arbitrary in the choice of the nodes and the edges that better embody prior knowledge. Nevertheless, we reasoned that these models could represent a testable framework whose biological/clinical significance can be challenged by comparing their predictions with observed biological or clinical data. In the following sections we report a use case where patient-specific genomic data are fed to Boolean models and used to infer clinical data.

### 3.5. The FLT3-NPM1-DNMT3A Boolean Model

As a first test we asked whether model perturbations caused by mutations observed in the driver genes in patients’ genomes would change the model output to match clinical data. We focused on the NPM1-DNMT3A-FLT3 causal module. This comutation pattern was found as the most frequent in the 1540 patient genomes characterized by Papaemmanuil et al. Interestingly, it was observed that the oncogenic impact of the FLT3-ITD mutation was most severe in patients with concomitant NPM1 and DNMT3A mutations, indicating that the prognostic value of one gene may be significantly altered if another gene is comutated. 

We used an NPM1-DNMT3A-FLT3 causal network encompassing 21 protein nodes and 3 phenotypes (Figure 5a and Table A1) to build a Boolean model and we recorded its predictions when different genomic profiles were considered as input to define the initial state of the network and its dynamics. 

In silico simulations were performed for different genomic profiles, including loss of function of NPM1 and/or DNMT3A and/or gain of function of FLT3. In our Boolean model simulation, oncogenes and TSG were set to fixed values of 1 and 0, respectively. Each node of the network is associated to a Boolean expression, describing how the value of the node changes depending on the activities of the upstream regulatory nodes. Nonmutated genes were considered active or inactive depending on the activities of the upstream regulatory nodes. This approach enabled to generate eight “mutation-specific” models. To estimate the level of activation of each phenotype, we assumed that the activities of the nodes directly linked to a phenotype have an additive effect on the value of the phenotype. Hence, we estimated phenotype activation by adding up the contribution of the upstream activators and subtracting that of the upstream inhibitors. Thus, a phenotype is considered “most active” whenever all activator nodes are on and inhibitors off at a steady state (Figure 5). 

This strategy yielded mutation-specific models integrating different genetic backgrounds into a causal network. Boolean simulations lead to equilibrium states characterized by different activation of the three AML hallmark phenotypes (Figure 5b–h).

### 3.6. Predictive Power of Boolean Models

We next evaluated whether we could use these models to infer some clinical outcomes of patients carrying mutations in NPM1 and/or DNMT3A and/or FLT3. To this end we made use of the clinical information provided by Papaemmanuil et al. reporting the impact of somatic mutations on overall survival and we compared the mutation-specific hazard ratios with the predictions of our models. As a proxy for the predictive power of our model, we defined the “integrated network phenotype” score, which is calculated by subtracting from the activation value of the phenotype “proliferation” the values of “apoptosis” and “differentiation”. As shown in Figure 6 (panels a–d), the phenotype scores correlate significantly with the death hazard ratio; the integrated network phenotype score shows the highest correlation. These conclusions were not affected by repeating the simulations with a model where the logic gates were based on an “activator wins” hypothesis. We next investigated the prognostic power of our model by comparing it with additional clinical features derived from the AML TGCA dataset. Specifically, we compared the mutation-specific peripheral blood (PB) and bone marrow (BM) blast percentages with the predictions of our models. As shown in Figure 6 (panels e and f), the integrated network phenotype score correlates nicely with the peripheral blood and bone marrow blast percentages (panels e–f). Additionally, as previously observed in the comparison with the death hazard ratio, the integrated network phenotype score shows the highest correlation as compared with single phenotypes (Figure 6g). Thus, we conclude that the prior experimental information embodied in our models is sufficient to replicate important clinical readouts in patients with different genomic profiles. 

## 4. Discussion

The rational design of drugs to cure complex diseases requires an understanding of the intricate crosstalk between genes whose mutations cause or modify the disease phenotype. This would allow the development of computational models to help infer the consequences of perturbing any given disease gene by candidate drugs. Modeling the molecular pathways that are perturbed in a polygenic disease, however, poses challenges. On the one hand, cell physiology is governed by a large network of interactions between many thousands of gene-products, a network whose details are often poorly understood. The assembly of a “complete” model faithfully reproducing cell physiology and pathology is beyond our current capabilities. Thus, approximate approaches should be pursued. On the other hand, setting boundaries delimiting the relatively isolated areas of the protein interaction network that are relevant to model a disease is hard. Over the last decades, network-based approaches have been applied to elucidate the molecular mechanisms underlying complex diseases [23,24].

In some cases, these strategies have been successfully employed to stratify patients and to identify new promising potential therapeutic targets in cancer [25,26,27]. 

Assembling a disease-relevant predictive network is not straightforward and different approaches have been proposed [28]. Some of these are unbiased while others are based on prior knowledge and rely on expert decisions. Reverse engineering approaches allow researchers to draw networks in an unbiased manner by using genomewide gene expression data to infer relationships between genes [29]. To make it simple, if two genes are coexpressed they are inferred to be functionally correlated and are linked in a gene regulatory network. The network resulting from these unbiased approaches is useful for building scaffolds when little information is available on the biological problem under study but is not appropriate when the goal is that of obtaining logic models to be used in Boolean simulations. In addition, reverse engineering approaches rely on genomewide expression studies that provide information for determining gene regulatory networks but say little about signaling networks where protein modification and modulation of stability play an important role that cannot be inferred from genomewide transcriptomics. Another approach consists of identifying genes that are frequently mutated in a disease and linking them, taking advantage of prior knowledge. The method builds an ensemble of logic-based dynamic models and trains them to experimental perturbations. The predictions of the model ensemble are finally combined into an ensemble prediction [24,30]. Networks solely based on physical interactions, however, miss an important piece of information as they do not consider the functional consequences of the interactions (activation, inhibition). To our knowledge, only a few network-based studies take advantage of causal information [10,19,31,32]. This could be explained by the limited information coverage on causal relationships as compared to physical interactions. Importantly, the value of causal interactions in network-based approaches has now been widely recognized and recently new resources annotating these types of relationships have been developed [11,31]. 

Here, we propose a novel—generally applicable—network-based strategy to obtain predictive logic models inferring relevant patient-specific clinical features. Our four-step strategy is based on the combination of expert curation with bioinformatics tools developed in two resources, SIGNOR and CancerGeneNet [11,12]. A catalogue of genes associated with a given disease represents the first step of our strategy. We next take advantage of the causal relationships annotated in the SIGNOR database to connect these disease-specific genes and to link them to key phenotypes, e.g., cancer hallmarks, to automatically obtain a “disease-specific network”. However, as in complex diseases the number of disease genes tends to be large, the network assembled by this approach is often too complex and unsuitable for Boolean modeling. In addition, it may contain relationships that are not relevant in the pertinent biological context. For these reasons, we recommend applying bioinformatics and “expert curator” filters to obtain a compressed disease-specific network, which is more suitable for logic modelling.

One concern regards the “expert curator” filter, which gives our approach a certain degree of arbitrariness as it is left to the curator to decide which of the possible relationships are most functionally relevant. In principle, this step could be avoided if context-specific weighed directed graphs were available. We anticipate that in the near future the emerging growth of omic data will enable researchers to give a “context-dependent score” to each causal relationship. This would contribute to making our strategy unbiased. As a final result, we aim to obtain a network that is a compromised between coverage and simplicity and is intended as a functional framework to help in diagnosis and therapy-decisions.

As a use case, here we applied our strategy to acute myeloid leukemia, a complex and heterogeneous cancer impacting the regulation of the hematopoietic differentiation process [32]. Our approach generated a signed directed “AML network” recapitulating literature-derived molecular mechanisms linking AML cancer driver genes to cancer hallmarks. Here we show that by combining patient-specific mutation profiles with the AML network, we obtain actionable Boolean models that enable to infer how genetic perturbation of a node impacts the cancer hallmarks. These assembled networks, together with the annotation of the experimental evidence supporting each relationship, can be visualized and downloaded from the CancerGeneNet resource for local use (https://signor.uniroma2.it/CancerGeneNet/). Importantly, these networks should not be considered as established definitive descriptions of the disease, but rather as continuously updatable models, whose predictions should be challenged with new experimental findings. 

Our results show that already in its present form the model can infer, with good accuracy, whether any of the nodes mutated in the tumor can be classified as an oncogene or a tumor suppressor, as the automatic prediction is largely in accord with expert annotation. In addition, a Boolean model derived from a smaller network-module representing the crosstalk between FLT3, DNMT3A, NPM1 and cancer hallmarks, when primed with patient-specific genomic profiles, yielded predictions that are in accord with patients’ clinical data.

Although successful, our strategy has some limitations that should be addressed in the near future. The module models that we have developed only aim at capturing the contribution of cancer driver genes to the development of cell phenotypes. Additional expert curation would be required to produce every subnetwork recapitulating each patient mutation profile. The long-range effects from modifier genes are not considered. In addition, cell phenotypes are not only determined by the mutational profiles, as environmental perturbations may have an impact on clinical phenotypes. These long-range effects and epigenetic contributions, however, could be captured from the analysis of perturbation of patient expression profiles, which are becoming increasingly more available in clinical settings. In a few cases, these profiles have already demonstrated some value in predicting clinical outcomes [33,34,35,36]. As gene expression data are modulated both by genomewide genetic and epigenetic information, patient-specific gene expression profiles should be overlaid onto the Boolean models, for instance by priming the model initial state This might contribute to tune the model predictions. Our strategy addresses the problem of assembling a disease network by exploiting a resource of annotated causal protein interactions. We have also shown that the generated networks can be turned into patient-specific actionable Boolean models that predict clinical outcomes. Logic modelling has already been used to model cancer pathways [37,38]. Others have used publicly available causal models and have challenged their predictions with perturbation data to obtain context-specific models [15,39,40]. The PROFILE method integrates mutation data, copy number alterations and expression to obtain patient-specific models. The novelty of our approach consists in the combination of disease-specific causal networks with patient-mutation profiles.

Finally, here we have demonstrated that minimal models that only consider the causal interactions between a few gene products can help rationalize and possibly infer relevant diagnostic and prognostic patient-specific features. 

The strategy that we have proposed is generally applicable and can be used to integrate tumor mutation profiles into molecular networks that are both biologically and clinically informative.

## Figures and Tables

**Figure 1 jpm-11-00117-f001:**
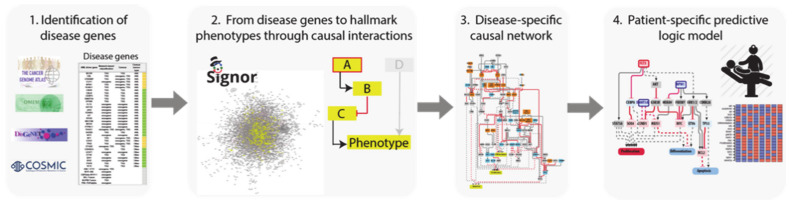
Schematic representation of the network-based strategy.

**Figure 2 jpm-11-00117-f002:**
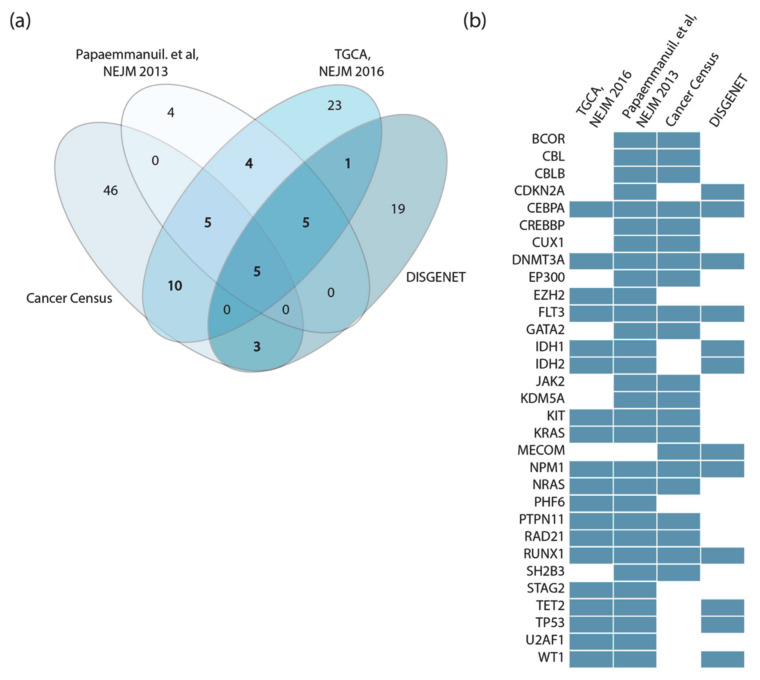
Acute myeloid leukemia (AML) cancer driver genes. (**a**) Venn diagram illustrating the overlap of the AML driver genes annotated in four independent resources and datasets. (**b**) Heatmap showing the 31 AML driver genes identified (blue rectangles) or not identified (white rectangles) in each indicated dataset. Only genes annotated in at least two resources are shown.

**Figure 3 jpm-11-00117-f003:**
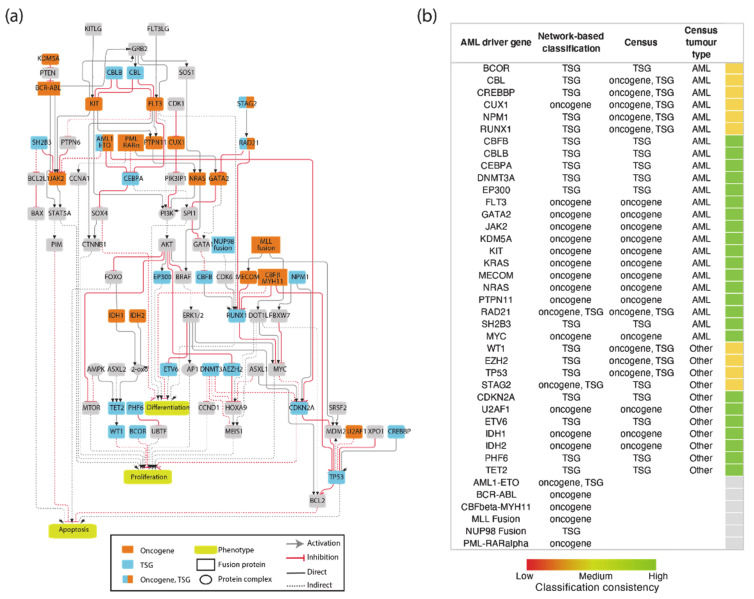
The AML causal network. (**a**) AML drivers are connected in a signed directed logic network to the three hallmark phenotypes. Each node and edge is color-coded as described in the legend. (**b**) Classification of AML driver genes as oncogenes or tumor suppressors based on inference from network connectivity and comparison with the annotation of the Cancer Gene Census.

**Figure 4 jpm-11-00117-f004:**
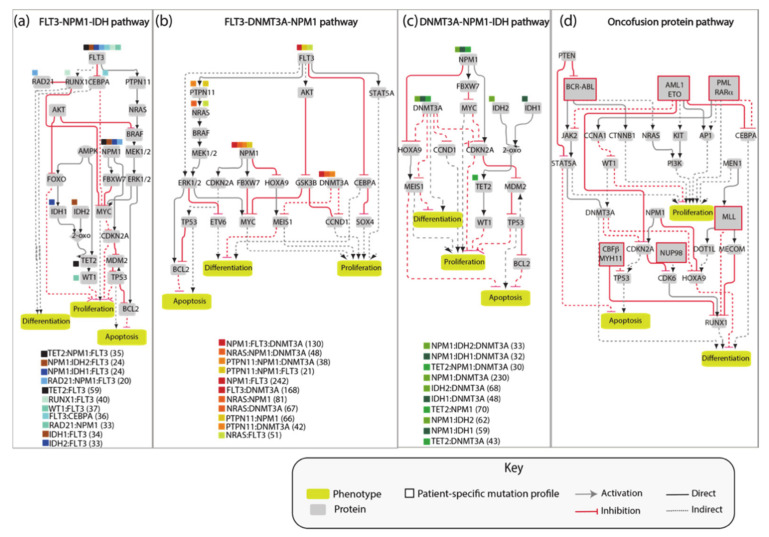
AML modules. Schematic of four AML-specific pathways recapitulating the mechanisms modulated by the most frequently identified co-occurring mutations (**a**–**c**) and onco-fusion proteins (**d**). For each of the three graphs in (**a**–**c**), the gene combinations that are frequently found mutated in patients are listed below the graph and assigned a colored square. In parenthesis are the number of occurrences in the cohort of 1540 AML patients [8]. The colored squares, assigned to the gene combinations, are used in the graphs to label the co-occurring genes.

**Figure 5 jpm-11-00117-f005:**
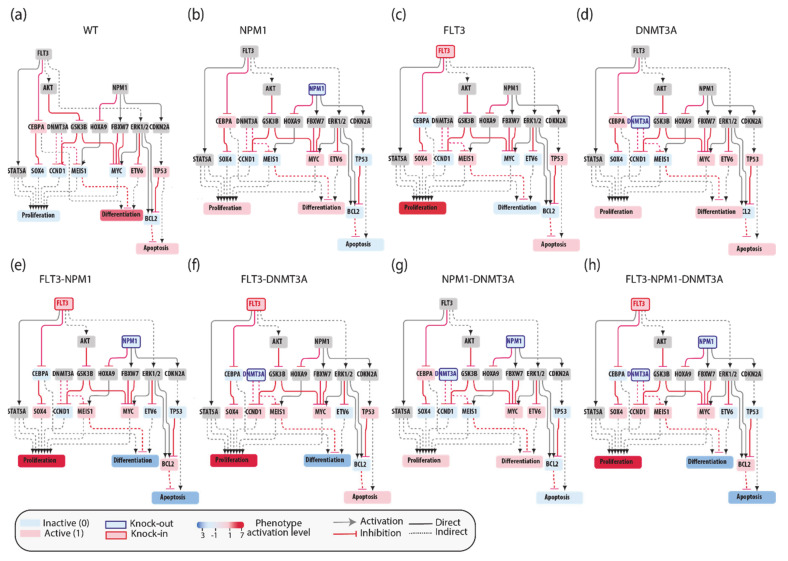
Boolean network simulations of the FLT3/DNMT3A/NPM1 module. In silico simulations of different genomic profiles, including wt (**a**), loss of function of NPM1 and/or DNMT3A and/or gain of function of FLT3 (**b**–**h**). The nodes are color-coded according to activation level at equilibrium.

**Figure 6 jpm-11-00117-f006:**
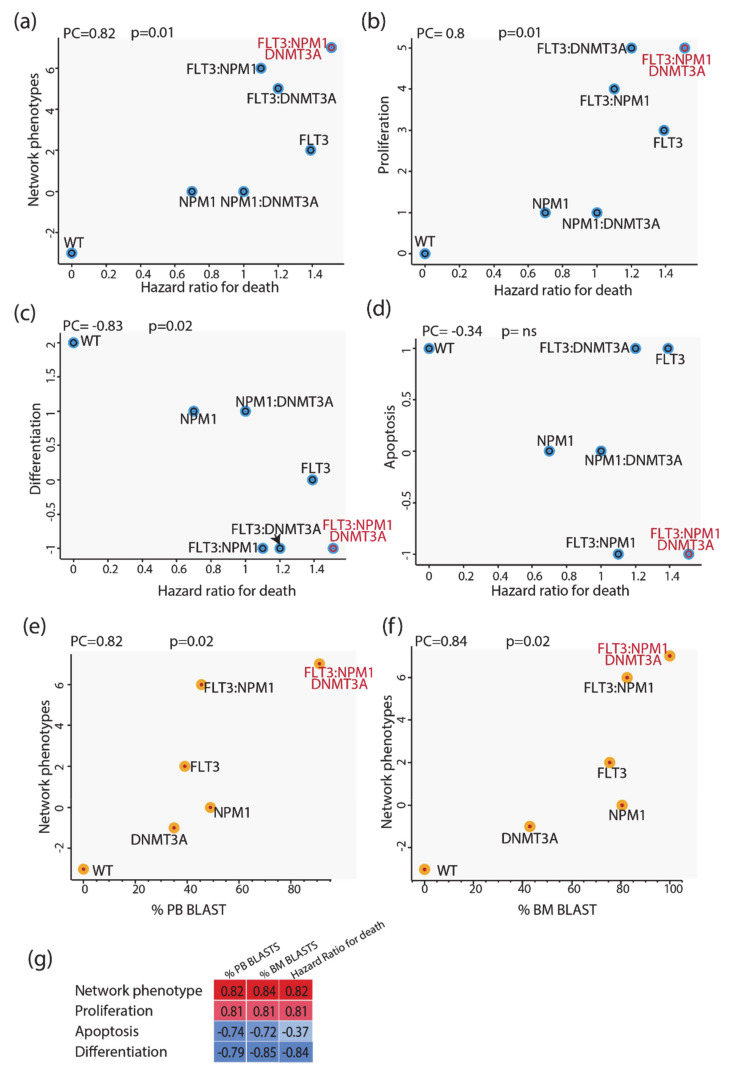
Predictive power of Boolean models. Scatterplot comparing the mutation-specific hazard ratios with the predictions of our models, as revealed by the network-score (**a**) and integrated network phenotypes-score (**b**–**d**). Scatterplot comparing the mutation-specific peripheral blood (PB) and bone marrow (BM) blast percentages with the integrated network phenotypes-score (**e**–**f**). (**g**) Heatmap showing the Pearson Correlation (PC) between clinical features (columns) and network phenotype scores (rows).

## Data Availability

The data presented in this study are openly available at signor.uniroma2.it/CancerGeneNet. Some of the results in this report are in part based upon data generated by the TCGA Research Network: https://www.cancer.gov/tcga.

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
