# Peer review of "Integrating Patient-Specific Information into Logic Models of Complex Diseases: Application to Acute Myeloid Leukemia"

_jpm, 2021, doi:10.3390/jpm11020117_

Round 1
Reviewer 1 Report
In the paper “Integrating patient specific high-content information into logic models of complex diseases: application to Acute Myeloid Leukemia”, Iannuccelli et al. developed a network-based strategy to explain possible Acute Myeloid Leukemia (AML) phenotypes starting from DNA mutational data. They integrated two bioinformatic resources (SIGNOR and CancerGeneNet) with “expert curator” filters to obtain predictive logic models inferring relevant patient-specific clinical features. They put genes frequently mutated in AML in the causal network generated by bioinformatic tools and they predict cancer pathways (proliferation, differentiation, apoptosis) activated/inhibited by different mutations (alone or together).
In order to reduce network complexity, they generated 4 sub-networks recapitulating the mechanism modulated by the most frequently co-mutated genes in 1540 AML patients characterized by Papaemmanuil et al.
The idea is interesting. They tried to simplify a complex and heterogeneous disease such as AML using a Boolean model that resemble clinical decision trees used by clinicians.
My doubts are the high degree of arbitrariness in the methods that do not permit to repeat paper results by other scientists, the lack of a strong validation of results and the excessive simplification of the original network that could not explain well all patients analyzed. In the era of Machine learning, maybe a Boolean approach could be too simplistic. Several genes mutated in AML (DNMT3A, TET2, IDH1/2, etc.) are epigenetic modifiers and they are not simple switches that activate/inhibit a certain signaling pathway but affect several pathways at the same time.
The paper is well written but sometimes the comprehension is more difficult because some network features explained in the results were not well explained in methods section.
My comments are:
- In my opinion the degree of arbitrariness in this approach is high. Your results are difficult to repeat by another scientist. There are several expert-curator decisions that are not well explained in the methods. There is, also, the lack of statistical test or data simulation to understand if the curator choices were correct. In particular: 1) The choice of initial driver genes it is fundamental to create the backbone network. Why did you choose these heterogenous datasets to choose driver AML genes? They used different platform to identify variants (WGS, WES, gene panels), different patients recruitment, different variants oncogenic potential definition. 2) Which papers did you use to connect driver genes to hallmark? Which are the key words used to identify them? Why did you choose only proliferation, differentiation and apoptosis? 3) The network cut and prune part could be very different if performed by another expert and subnetworks were reduced arbitrarily using less information than the original network; 4) you used the arbitrary “inhibitor wins” approach for combining the effect of different input. Did you test your model also with an activator wins approach in order to understand if your validation shows opposite results?
- One of your results is that this model can infer, with good accuracy, whether any of the node mutated in the tumor can be classified as an oncogene or a tumor suppressor. In section 3.4 you affirmed “Genes that are connected to the phenotype proliferation only by activating edges or by an even number of inactivating edges would be classified as oncogenes, alternatively, as tumor suppressors. A similar strategy was used to classify genes impacting differentiation or apoptosis. This approach enabled us to classify 18 genes as oncogenes and 19 as tumor suppressor”. How did you exactly classify genes as oncogenes? Did you take in account only proliferation classification or all phenotypes? What did you do when there was discordance by classifications obtained in different phenotypes? Genes connected only by activating edges to apoptosis were classified as oncogenes or tumor suppressor? And differentiation? Please explain better your classification rules.
- I do not understand very well the rules to set initial network conditions. How did you establish if a mutation produces gain or loss of function? Did you use previous classification (oncogene or TSG) to choose mutation effects and to set a value (0 or 1) to each node? If an oncogene is mutated is considered as constitutive oncogene or its effect is reverted? And for tumor suppressor genes? In the model, for instance, if a TSG usually activate another gene and it is mutated, it represses this gene? Non-mutated genes were considered as functional active? Please clarify in the text.
- Subnetworks are useful to understand the model. But they were not complete to analyze all samples. Did you test the samples of figures 6 and 7 in the whole network (the one depicted in the figure 4)? Are the results comparable? Please make this comparison.
- To validate your model you used data from Papaemmanuil paper that was used also to construct the model. Did you use also TCGA data in order to compare your prognostic powerful with the other based solely on genomic profiles showed in figure 1? I think it could more useful use also an external cohort in order to confirm the robustness of your approach. If another dataset exists, please verify your prognostic powerful.
- In the figure 7 you introduce the network-score. Did you refer to “the combined effect of the three phenotypes, by subtracting to the activation value of the phenotype “proliferation” the values of “apoptosis” and “differentiation”? Please add network score definition to the text.
- In figure 7, please add the number of samples for each group and the statistical significance of correlation value obtained.
- Your model is based on a gene pathways network. Moreover, you decided that activation of proliferation has an oncogenic effect while activation of differentiation and apoptosis a onco-suppressive effect. Why you did not check if mutations (or a combination) affect gene expression related to proliferation, differentiation and apoptosis? If a dataset with genomic and transcriptomic data exists, please validate your approach with it.
- Why did you use overall survival to validate your model? Survival could be changed by therapies and treatment. For instance, if you feed your model with an Acute promyelocytic leukemia patient with PML/RARA fusion gene, you’ll probably find the activation of proliferation and the blockage of differentiation while its survival will be good because ATRA treatment save more than 90% of patients.
- I think that results in section 3.1 are not very useful for the development of the model and it could be removed. You affirmed: “ (in TCGA data) we observed little correlation between patient cytogenetic risks and overall survival across genomic/transcriptomic subgroups obtained by unsupervised clustering”. It is well known that cytogenetic risk used in TCGA samples that were based only on cytogenetics data did not explain the complexity of prognosis and Papaemmanuil exposed this concept in her paper. Morover, you affirmed that genomic/gene expression profiles have uncertain prognostic value. In my opinion your analysis made with TCGA transcriptomic data is not properly correct because you used all the 19,000 genes to cluster patients. In these genes there are housekeeping genes, not expressed genes, and low-expressed genes that creates too much noise for clustering algorithm (which algorithm? Which distance? It is not specified in the text). Did you make this analysis using most variable genes? Moreover, several gene-expression prognostic signatures have been developed from TCGA data suggesting a possible prognostic value of transcriptomic data. At the same time Papaemmanuil developed a new method to improve classical prognostic classification using only genomic data.
- In figure 6. In panels a and d how is possible that MYC, which is regulated only by FLT3 (through AKT-GSK3B and ERK1/2) and NPM1 (through FBXW7), has discordance in activation? Please, check this problem.
- In the text, I found some citations which did not reflect the sense of the sentences and it makes more difficult to understand the paper. In section 3.3 referring to genomic data in AML, “These data are annotated in a number of resources and publications [16]”, but the paper talk about BoolNet R package. In section 3.4 “This automatically generated network was further reviewed and pruned, taking into account expert consensus as extrapolated from highly cited reviews [22, 23] to generate a simpler graph of 81 nodes”. 22 and 23 are not reviews but are application of Boolean models, are the citations correct? Please clarify your references.
Reviewer 2 Report
This paper proposes a computational method to model complex diseases like Acute Myeloid Leukemia (AML) by embedding patient specific genomics data into actionable logic models with prior knowledge. The authors first assembled a logic network with most AML-related mutated genes, which were then integrated with literature-derived causal networks and patient-specific high content. This paper is interesting and worth investigating. However, there are some concerns that need to be addressed to enhance the quality of the paper.
- The method part is not clearly presented. While the authors simply describe some probable components of the method like casual info, network and boolean model, it is unclear how each of these method parts are correlated. Moreover, the authors further described "a network-based strategy" in the Results section, which further made the structure of the method descriptions confused. The authors need to reorganize the structure of the paper to better reflect how each of the components of the method work with each other. Preferably, a workflow and more detailed elaborations on the general method should be provided.
- While the curation of the causal information in Section 2.1 seemed to make sense, a more reasonable way is to use some unbiased models (such as machine learning models) to select related causal features from a larger pool of candidate genes with unbiased models. In this case, the authors could avoid neglecting other important genes or including irrelevant genes. For example, a LASSO model [1-2] or an elastic net model [3-4] can select important features from a large number of features which may be organized in a network. The authors should discuss the advantages or disadvantages of their model compared with these unbiased models.
[1] Wan, S., Mak, M. W., & Kung, S. Y. (2015). mLASSO-Hum: A LASSO-based interpretable human-protein subcellular localization predictor. Journal of theoretical biology, 382, 223-234.
[2] Lu, Y., Zhou, Y., Qu, W., Deng, M., & Zhang, C. (2011). A Lasso regression model for the construction of microRNA-target regulatory networks. Bioinformatics, 27(17), 2406-2413.
[3] Sokolov, A., Carlin, D. E., Paull, E. O., Baertsch, R., & Stuart, J. M. (2016). Pathway-based genomics prediction using generalized elastic net. PLoS computational biology, 12(3), e1004790.
[4] Wan, S., Mak, M. W., & Kung, S. Y. (2015). Mem-mEN: predicting multi-functional types of membrane proteins by interpretable elastic nets. IEEE/ACM transactions on computational biology and bioinformatics, 13(4), 706-718. - I would prefer to see more quantitative measurements, such as accuracy, stability, etc, about the proposed method on modeling the AML disease. The accuracy tells how well the model performs on the AML, and the stability tells whether the proposed model can be extended to modeling other complex diseases.
- The authors should compare their method with state-of-the-art methods to demonstrate the feasibility and superiority of the proposed method. The comparison results should be reported.
- It is not clear how the patient-specific high-content information contributed to the success of the model. The authors should emphasize and elaborate this contribution in a depth discussion.
- The authors used the so-called causal information in the network. In a network modeling, that means the network is directed. However, I do not see any place in the manuscript emphasizes this and how the directed network was modeled and used to do some prediction such as clinical outcomes. The authors should clarify this.
Round 2
Reviewer 1 Report
In the revised paper “Integrating patient specific high-content information into logic models of complex diseases: application to Acute Myeloid Leukemia”, I appreciated that the authors answered to my comments.
In particular, I appreciated the comparison between “inhibitory wins” and “activator wins” simulations and the clarification of methods procedures and definitions.
I have some doubts about other answers:
- “The relatively high number of nodes made the calculation of the attractors computationally expensive in terms of time and memory. Thus, we couldn't complete the simulation by using the BoolNet package on our hardware. This was the reason behind our decision to build simpler Boolean models appropriate for each question”. In this context, your model needs additional “expert curation” to produce every subnetwork for each patient mutation profile. Moreover, if a model is too much computationally expensive, maybe the approach is not the best for its purpose.
- In my opinion, validation results showed a strong influence of proliferation pathway (and its “opposite” pathway differentiation) biased by FLT3 effect while apoptosis, which is a more complex biological event, had no significant correlation with survival. Percentage of blasts is an end point influenced mainly by proliferation. Moreover, this percentage is one of the major prognostic factors in AML. In this context, it seems that your model is able to calculate mainly this aspect of AML which doesn’t respect the heterogeneity of the disease. Please use other genes where the proliferation power is less important than FLT3.
I have got other doubts about the aim of this study:
- Which is the aim of the study? Because if you want to show an innovative tool, you must make it available to users, without doubts about arbitrariness and easy to use. While, if you want to present an application of your tool you have to show new discoveries in AML field and in my opinion your results were not very innovative.
- Who could be a user for your tool? A clinician? A molecular biologist? If a clinician uses your tool for a patient finding a strong proliferation network value, how can select/change the patient treatment? Using broad anti-proliferation drugs? In the era of personalized medicine, I think is an anachronistic approach. If a molecular biologist finds the same results, he/she will try to develop new anti-proliferative drugs? Please clarify the aim of your paper.
Author Response
Reviewer 1
In the revised paper “Integrating patient specific high-content information into logic models of complex diseases: application to Acute Myeloid Leukemia”, I appreciated that the authors answered to my comments.
In particular, I appreciated the comparison between “inhibitory wins” and “activator wins” simulations and the clarification of methods procedures and definitions.
I have some doubts about other answers:
- “The relatively high number of nodes made the calculation of the attractors computationally expensive in terms of time and memory. Thus, we couldn't complete the simulation by using the BoolNet package on our hardware. This was the reason behind our decision to build simpler Boolean models appropriate for each question”. In this context, your model needs additional “expert curation” to produce every subnetwork for each patient mutation profile. Moreover, if a model is too much computationally expensive, maybe the approach is not the best for its purpose.
In our work we aimed at building models sufficiently simple to be computationally manageable while at the same time containing most of the crosstalk information to reproduce experimental results and infer clinically relevant outcomes. We have shown that this approach applied to patient genomes with different combinations of mutations in the FLT3 NPM1 DNMT3A genes delivers meaningful clinical predictions. Boolean simulations of models with hundreds of nodes are feasible with a hardware setup of moderate power. We couldn’t do these simulations. For what it’s worth, we are confident that the simulation proposed by the referee on the large AML network will have given results similar to those of the smaller module on a workstation with sufficient RAM. On the other hand, AML is a heterogeneous disease and building a one-fits-all model might be a worthless effort. This to justify our approach of splitting the large model into smaller ones, each one fitting different genetic context.
Although we think that additional expert curation would be extremely useful to obtain subnetworks recapitulating each patient mutation profile [1], we would like to highlight that our model was able to infer clinical features of patients harboring different combination of mutations in the FLT3, NPM1 and DNMT3A genes. The referee suggestion prompted us to add one sentence in the discussion about this point.
- In my opinion, validation results showed a strong influence of proliferation pathway (and its “opposite” pathway differentiation) biased by FLT3 effect while apoptosis, which is a more complex biological event, had no significant correlation with survival. Percentage of blasts is an end point influenced mainly by proliferation. Moreover, this percentage is one of the major prognostic factors in AML. In this context, it seems that your model is able to calculate mainly this aspect of AML which doesn’t respect the heterogeneity of the disease. Please use other genes where the proliferation power is less important than FLT3.
The analysis of the referee is correct. FLT3 is a pro-proliferative gene while the status of the apoptosis phenotype does not seem to correlate with survival. However, we would not consider these remarks as a detraction for the model. It simply means that if we removed the CDKN2A, the gene that modulates apoptosis in our model, from the model we wouldn’t loose much of the predictive power in this genetic context. However, we considered this gene in the model as CDKN2A is a tumor suppressor that is found mutated in some AML patients and its impact on apoptosis might be useful to rationalize links with survival in more complex genetic backgrounds
I have got other doubts about the aim of this study:
- Which is the aim of the study? Because if you want to show an innovative tool, you must make it available to users, without doubts about arbitrariness and easy to use. While, if you want to present an application of your tool you have to show new discoveries in AML field and in my opinion your results were not very innovative.
The aims of the study are:
- to review the literature on the molecular defects in AML and present their interactions in a network representation that could reveal how the observed gene products and their mutations crosstalk to impact AML hallmark phenotypes.
- to propose a network-based strategy. As we mentioned in the discussion section, most of the network-based strategies rely on physical interactions. Here the novelty consists in using causal interactions to link disease-specific genes and obtain a “Causal disease-specific network”. We show that this network can represent a rational platform to explain complex phenotypes in a heterogeneous complex disease. In addition, we demonstrate that derived actionable Boolean models that are consistent with published findings and in specific genetic contexts predict clinical outcomes that were not used to build the models.
One new contribution, perhaps not “discovery”, is the molecular explanation of the cross talk between FLT3 NMP1 and DMT3A leading to the cooperative effect of the three mutations in determining to the severity of the disease.
- Who could be a user for your tool? A clinician? A molecular biologist? If a clinician uses your tool for a patient finding a strong proliferation network value, how can select/change the patient treatment? Using broad anti-proliferation drugs? In the era of personalized medicine, I think is an anachronistic approach. If a molecular biologist finds the same results, he/she will try to develop new anti-proliferative drugs? Please clarify the aim of your paper.
Granted that we are not yet offering a clinical tool to direct therapeutic decisions, we envisage that a clinician by inspecting our AML network and the perturbations in the network caused by the patient specific mutations altered could identify the signaling path(s) that link the mutated gene(s) to the observed phenotype may, in principle, design a patient-specific strategy that activates or represses a node along the path. Mind it, we do not claim that our network should be used as it is as a tool in clinical settings for automatic decisions or choice of therapeutic strategies. We do not claim that we are offering a tool to diagnose patient from genomic profiles. We only show that a Boolean model based on our network reproduces clinical outcomes when fed with information on patient genomic profiles. Starting from this, when we will observe that the model prediction contrasts with clinical results, we will use this contradiction to refine the model.
References
- Henriques, D., et al., Data-driven reverse engineering of signaling pathways using ensembles of dynamic models. PLoS Comput Biol, 2017. 13(2): p. e1005379.
- Pe'er, D. and N. Hacohen, Principles and strategies for developing network models in cancer. Cell, 2011. 144(6): p. 864-73.
- Stolovitzky, G., D. Monroe, and A. Califano, Dialogue on reverse-engineering assessment and methods: the DREAM of high-throughput pathway inference. Ann N Y Acad Sci, 2007. 1115: p. 1-22.
- Barabasi, A.L., N. Gulbahce, and J. Loscalzo, Network medicine: a network-based approach to human disease. Nat Rev Genet, 2011. 12(1): p. 56-68.

Reviewer 2 Report
The authors tried to address the concerns. However, not all concerns were fully addressed.
- First and also one of the most important point, is to mark (either in different colors or in bold style) those newly changed or added sentences, paragraphs in the revised manuscript for ease of checking. Please do so for all the responses to all comments in the first round of review.
- For the first comment of the last-round review, I did not see clearly how the authors "clarify our network-based strategy". Please either mark them in the revised manuscript or copy those sentences in the response.
- For the second comment, again I did not see the so-called "discussions in page 10". Besides, I would like to emphasize that this comment was concerning the possibility of missing other important genes potentially due to not covering all literature or because of subjectively selecting genes as different people might have different understandings of some genes to determine whether to include or exclude them. That's why some references with unbiased approaches were mentioned to compare. In this case, please provide evidence to suggest "The network resulting from these unbiased approaches are usually very complex and dense of 'noisy' interactions, they provide very little mechanistic insight and are difficult to interpret." These are very serious claims which require substantial evidence to support; otherwise, the authors should discuss qualitatively (if quantitatively not possible) the advantages and disadvantages by comparing the proposed method against these unbiased approaches.
- For Comment 4 of the last-round review, as the authors suggested that they used some strategies to establish a feasible approach, the authors can either compare some methods without one or more strategies they proposed to demonstrate the necessity of each proposed strategy. For example, the authors integrated patient specific information into the models. What about those models without this information, or models with information but not patient-specific? There are always a bunch of different models to compare, depending on what particular contributions of this manuscript the authors would like to emphasize.
- For other comments, please mark those changed parts in the revised manuscript and also specify where you made the changes in the response.
Author Response
Reviewer 2
The authors tried to address the concerns. However, not all concerns were fully addressed.
- First and also one of the most important point, is to mark (either in different colors or in bold style) those newly changed or added sentences, paragraphs in the revised manuscript for ease of checking. Please do so for all the responses to all comments in the first round of review.
We apologize for not uploading a track-change version of our manuscript. In the new revision the changes in revision 1 are in red while the new one are in ‘trach-changes’.
- For the first comment of the last-round review, I did not see clearly how the authors "clarify our network-based strategy". Please either mark them in the revised manuscript or copy those sentences in the response.
As the reviewer can now see from the marked document, we had added a paragraph in the methods section. In addition, please consider that Figure 1 graphically illustrates the strategy and that the revised bullet points at page 4 also is meant to schematically clarify what is all about.
- For the second comment, again I did not see the so-called "discussions in page 10". Besides, I would like to emphasize that this comment was concerning the possibility of missing other important genes potentially due to not covering all literature or because of subjectively selecting genes as different people might have different understandings of some genes to determine whether to include or exclude them. That's why some references with unbiased approaches were mentioned to compare. In this case, please provide evidence to suggest "The network resulting from these unbiased approaches are usually very complex and dense of 'noisy' interactions, they provide very little mechanistic insight and are difficult to interpret." These are very serious claims which require substantial evidence to support; otherwise, the authors should discuss qualitatively (if quantitatively not possible) the advantages and disadvantages by comparing the proposed method against these unbiased approaches.
The text that was added is as follows (actually at page 11, sorry!). In blue are some addition for further clarification.
Assembling a disease-relevant predictive network is not straightforward and different approaches have been proposed [2]. Some of these are unbiased while others are based on prior knowledge and rely on expert decisions. Reverse engineering approaches allow to draw networks in an unbiased manner by using genome-wide gene expression data to infer relationships between genes [3]. To make it simple, if two genes are co-expressed they are inferred to be functionally correlated and are linked in a gene regulatory network. The network resulting from these unbiased approaches are useful to build scaffolds when little information is available on the biological problem under study but are not appropriate when the goal is that of obtaining logic models to be used in Boolean simulations. In addition, reverse engineering approaches rely on genome wide expression studies that provide information for determining gene regulatory networks but say little about signaling networks where protein modification and modulation of stability play an important role that cannot be inferred from genome wide transcriptomics. Another approach consists in identifying genes that are frequently mutated in a disease and link them taking advantage of prior knowledge. The method builds an ensemble of logic-based dynamic models and trains them to experimental perturbations. The predictions of the model ensemble are finally combined into an ensemble prediction [1, 4].
- For Comment 4 of the last-round review, as the authors suggested that they used some strategies to establish a feasible approach, the authors can either compare some methods without one or more strategies they proposed to demonstrate the necessity of each proposed strategy. For example, the authors integrated patient specific information into the models. What about those models without this information, or models with information but not patient-specific? There are always a bunch of different models to compare, depending on what particular contributions of this manuscript the authors would like to emphasize.
The scope of our work consists in proposing models that are useful to predict the equilibrium state of a system in a specific genetic AML context. The model always requires information about the genetic background as this affect the dynamic behavior of the system. This information can be “all the node are wild type” as in panel a of Fig. 5. We cannot see how we can satisfy the request of the referee. Unless the referee wants to know how the models respond to perturbations caused by random mutations never observed in patients. However, this was addressed, we believe, by the analysis of network robustness (see response to point 3 in the former rebuttal).
- For other comments, please mark those changed parts in the revised manuscript and also specify where you made the changes in the response.
We have now marked the changes in the manuscript. In the previous point-by-point response, we have specified the pages where we revised the manuscript according to the reviewer suggestions.
References
- Henriques, D., et al., Data-driven reverse engineering of signaling pathways using ensembles of dynamic models. PLoS Comput Biol, 2017. 13(2): p. e1005379.
- Pe'er, D. and N. Hacohen, Principles and strategies for developing network models in cancer. Cell, 2011. 144(6): p. 864-73.
- Stolovitzky, G., D. Monroe, and A. Califano, Dialogue on reverse-engineering assessment and methods: the DREAM of high-throughput pathway inference. Ann N Y Acad Sci, 2007. 1115: p. 1-22.
- Barabasi, A.L., N. Gulbahce, and J. Loscalzo, Network medicine: a network-based approach to human disease. Nat Rev Genet, 2011. 12(1): p. 56-68.
